# Supporting Design to Develop Rural Revitalization through Investigating Village Microclimate Environments: A Case Study of Typical Villages in Northwest China

**DOI:** 10.3390/ijerph19148310

**Published:** 2022-07-07

**Authors:** Kai Xin, Jingyuan Zhao, Tianhui Wang, Weijun Gao

**Affiliations:** 1School of Architecture, Chang’an University, Xi’an 710046, China; xinkai@chd.edu.cn; 2Faculty of Environmental Engineering, The University of Kitakyushu, Fukuoka 808-0135, Japan; 3School of Building Services Science and Engineering, Xi’an University of Architecture and Technology, Xi’an 710055, China; 4iSMART, Qingdao University of Technology, Qingdao 266033, China

**Keywords:** village microclimate, numerical simulation, global climate, outdoor thermal comfort, rural revitalization

## Abstract

China has the largest number of villages in the world, and research on rural microclimate will contribute to global climate knowledge. A three-by-three grid method was developed to explore village microclimates through field measurement and ENVI-met simulation. A regression model was used to explore the mechanistic relationship between microclimate and spatial morphology, and predicted mean vote (PMV) was selected to evaluate outdoor thermal comfort. The results showed that ENVI-met was able to evaluate village microclimate, as Pearson’s correlation coefficient was greater than 0.8 and mean absolute percentage error (MAPE) was from 2.16% to 3.79%. Moreover, the air temperature of west–east road was slightly higher than that of south–north, especially in the morning. The height-to-width ratio (H/W) was the most significant factor to affect air temperature compared to percentage of building coverage (PBC) and wind speed. In addition, H/W and air temperature had a relatively strong negative correlation when H/W was between 0.52 and 0.93. PMV indicated that the downwind edge area of prevailing wind in villages was relatively comfortable. This study provides data support and a reference for optimizing village land use, mediating the living environment, and promoting rural revitalization.

## 1. Introduction

Global climate change has become a major issue of common concern to governments and the scientific community [1], and is also the most essential challenge faced by mankind today. The influence of climate change on humans is comprehensive, multiscale, and multilevel [2,3]. Related research has found that the air temperature of Earth is about 1.1 °C warmer than that it was in the late 1800s, and the past decade (2011–2020) has been the hottest period on record [4]. Moreover, climate change has caused a series of problems, such as reducing indoor and outdoor thermal comfort [5], endangering human health [6,7,8], aggravating air pollution [9,10], and increasing building energy consumption [11], etc. Besides, global climate change will accelerate the speed of photochemical reactions, increase the concentration of ozone in the near-surface atmosphere, and affect human health [12].

Facing the disasters brought by climate change, humans in cities have been exploring how to respond to global climate change since the 1970s [13,14,15]. Scholars have proved that urban green parks can improve the thermal environment of surrounding built-up areas and create a healthier living environment for citizens [16,17,18,19,20]. Scholars have also studied the impact of improvement in different tree layouts (clustered and equal interval) on human thermal comfort and outdoor microclimates [21]. Landscape planning can improve the local microclimate [22], and relevant studies have also pointed out that local microclimate changes can affect building energy consumption [23]. However, most of the research has been on cities, with few on villages. Ref. [24] indicated that the maximum values of air temperature in rural area will increase by 0.6 °C, 1.6 °C, and 3.8 °C in the 2030s, 2050s, and 2080s, respectively, in Singapore. In Australia, what village residents were concerned about were the economic problems of climate change and government policies [25]. Climate studies in rural areas have only been based on macro perspectives and government policies [13,26,27], but there is a lack of exploration of the mechanisms of the village microclimate.

The rural economy has been boosted and the original natural landscape of villages have been changed with rapid urbanization since China’s opening reforms from 1978 [26,28]. Disorderly expansion of the construction area of villages, the increase in building heights, and the expansion of impervious underlying surfaces [27] have undoubtedly caused ground-surface temperatures to rise due to the thermal properties of artificial surfaces being higher than natural surfaces [29,30]. This also increases the burden of global climate issues due to China owning the largest number of villages in the world, especially in the northwest. Therefore, research on village microclimates is of great significance.

Rural revitalization is a key policy that was recently proposed by the Chinese government for supporting rural sustainable development [31,32], which is improving the rural living environment and enhancing the rural economy. Nowadays, research related to rural revitalization focuses mainly on rural e-commerce development [33], idle rural residential land [34], land consolidation [35], protection and inheritance of ancient villages [36,37], and rural tourism [38,39]. Rural tourism has become an essential measure of rural revitalization; however, a more comfortable outdoor environment could attract more tourists. It is absolutely necessary to create a more comfortable living environment by exploring rural microclimates.

Based on the above research, the purpose of this paper is to supplement the mechanism gap of village microclimates and provide references for rural revitalization. Typical villages in the northwest were selected to investigate the village microclimate environment. Firstly, the accuracy of ENVI-met was verified based on on-site surveys through investigating typical villages in northwest China. Secondly, a three-by-three grid method was developed to evaluate village microclimate environments, dividing villages into nine equal zones, each with the same area. Thirdly, predicted mean vote (PMV) was used to assess outdoor thermal comfort. Finally, relevant references for rural tourism are provided by the results of this paper.

## 2. Materials and Methods

### 2.1. Study Site

Two typical villages were selected to investigate the microclimate in northwest China—Cai Jia (CJ) and Da Nihe (DN)—which are located in Xi’an, Shaanxi (Figure 1). The average air temperature is 27.7 °C, mean relative humidity is 59%, and average wind speed is 1.6–2.5 m/s from June to August (summer). Both the villages are located in Xi’an city (34°10′–34°27′ N, 108°59′–109°16′ E) and separated by about 50 km, and therefore we considered that meteorological conditions of the two villages were similar.

### 2.2. ENVI-Met Simulation

#### 2.2.1. Introduction of ENVI-Met

At present, the methods of studying microclimates mainly include field measurement and computer numerical simulation: specifically, field measurement is limited to fixed locations, and numerical simulations have become the most popular method. For simulation software, many scholars use computational fluid dynamics (CFD) and ENVI-met software; however, CFD is mainly for simulating the wind environment, and ENVI-met has become the main method for microclimate simulation, which is a computational dynamics model developed by Bruse (German scholars) [40,41,42]. ENVI-met has been used worldwide for environmental analysis and urban planning from the tropics to polar regions, and the software’s potential has been validated and its calculations verified in over 3000 scientific publications and independent studies [43]. Moreover, it has been validated in various climatic conditions based on the locations, also including Northwest in china [44,45,46,47,48]. ENVI-met is also an integrated three-dimensional non-hydrostatic model for simulating air–plant interaction on the ground, with horizontal resolution of 0.5 to 10 m, typical time of 24 to 48 h, and time step of 1 to 5 s. ENVI-met version 4.4.5 was selected for this paper.

#### 2.2.2. Architecture Model Setup

The model of DN village consisted of a 671,824 m^2^ area (length = 844 m, width = 796 m), with 211 × 199 horizontal square grids. the CJ village model domain consisted of a 296,100 m^2^ (630 × 470 m) area, which horizontally consisted of 160 × 120 square grids. The resolution of the horizontal direction of the two villages was 4 m per grid. This study used 10 vertical grids with a resolution of 2 m to make sure that the vertical height of the model is double that of the highest building in the village. Ten nest grids were added to move the boundary away from the model core and to minimize boundary effects to ensure the stability of the simulation. The same precision and nested grids were used in both villages to reduce simulation errors.

Input parameters of buildings and roads were determined from the topographic map of the village, and the type of underlying surfaces was defined through on-site interviews, mainly on: soil and concrete paving. However, local vegetation was not considered, because the landscape was dominated by uneven weeds and had few trees. Finally, the database from the current ENVI-met model was utilized, which supports thermal parameters of building materials and underlying surfaces for all structures, as illustrated in Table 1.

#### 2.2.3. Field Measurement and Meteorological Parameters

Four sites in the DN village were selected for field measurement, and measurement data were used to verify the accuracy of the ENVI-met model. Air temperature (AT) and relative humidity (RH) were simultaneously recorded and automatically stored with an interval of 30 min by a portable meteorological instrument (Kestrel 4500) and a portable temperature detector (TP49 AC.0 sensor model, accuracy of 0.25 °C). Field measurement was conducted between 8:00 and 23:00 from 20 to 23 July 2018, which is the hottest period in the year. Data on July 21 were chosen for this study, because the day was sunny and breezy, with daily average air temperatures higher than 30 °C.

The nearest weather station was referenced as the input meteorological parameter, as illustrated in Table 2, which is located in Xi’an city. Measurement value of AT and RH was used to verify the accuracy of the ENVI-met model between 8:00 and 23:00 on 21 July 2018, mainly as the measurement period was from 8:00 to 23:00. However, the total simulation time was set to 36 h in order to reduce errors.

### 2.3. Outdoor Thermal Comfort

Predicted mean vote (PMV) is an evaluation index proposed by Professor Fanger in Denmark to characterize the thermal response of the human body (hot and cold sensation) [49], which represents the average hot and cold sensation of most people in the same environment [50,51].

Using PMV as a thermal sensation evaluation index for outdoor thermal comfort has been proved in many studies, e.g., thermal comfort issues in an urban station square were investigated with the PMV thermal index in the Hokuriku region in Japan [52]. The thermal sensation in the open spaces was evaluated in the Faculty of Engineering, Assiut University, Egypt through PMV [53]. The accuracy of thermal sensation of PMV was verified through comparing 20 outdoor scenarios and 11 outdoor scenarios [54]. Thus, this paper selected PMV as the outdoor thermal comfort index to evaluate the village thermal environment. The equation of *PMV* is as follows:(1)$$\begin{array}{l} PMV=[0.303*e^{-0.036M}+0.028]\{M-W-3.05*10^{-3}[5733-6.99(M-W)-P_{a}]-0.42\\ {}[(M-W)-58.15]-1.7*10^{-5}M(5867-P_{a})-0.0014M(34-t_{a})-3.96*10^{-8}f_{cl}.[{\left(t_{cl}+273\right)}^{4}\\ -{\left(\bar{t_{s}}+273\right)}^{4}]-f_{cl}h_{c}(t_{cl}-t_{a})\} \end{array}$$

Parameters included in the *PMV* equation:*M*—Metabolic rate, W/s;*W*—Work Metabolism, W/s;*P_a_*—Partial pressure of water vapor in ambient air, Pa;*t_a_*—Air temperature, °C;*f_cl_*—The ratio of the surface area of the clothed body to the naked body;*t_s_*—Mean radiant temperature, °C;*t_cl_*—Human body surface temperature, °C;*h_c_*—Convective heat exchange coefficient, W/s·m^2^·°C.

Personal human parameters are derived from the ISO7730 standard (Table 3), which is the most appropriate method at present, although the mannequin cannot represent all inhabitants [55,56]. According to previous studies, PMV range can be extended from very cold to very hot (−4, +4), with 0 representing a neutral state [52]. However, the PMV can be above and below the range (−4, +4) as it is a mathematical function of the local climate during most literature applications [55].

### 2.4. Data Analysis

Nine equal zones were divided through the method of three-by-three grids to evaluate the mean village microclimate environment. Figure 2a illustrates this method.

Studies have shown that a buffer zone with 150 m around is an effective distance to affect local microclimate [57], similar to what used to this paper, as illustrated in Figure 2b, which is to draw a circle with 150 m as the diameter based on the center of each zone [58]. The percentage of building coverage (PBC) for each circle was calculated with AutoCAD through the village map. The calculation method of building coverage percentage was: PBC = building area/π·*R*^2^, *R* = 150 m. The H/W (height-to-width ratio, which is the ratio of building height to road width) was also used as a spatial morphology indicator to explain the relationship of village microclimate. Table 4 shows the PBC and H/W in the study area.

The AT of vertical 1.5 m from the roads of south–north (S-N) and west–east (W-E) were used as the microclimate evaluation index. PMV was used as the human thermal comfort evaluation index. The coefficient of determination (*R*^2^), Pearson’s correlation coefficient, and mean absolute percentage error (MAPE) were chosen to verify the accuracy between simulation and measurement, and for the accuracy of the ENVI-met model. Although the simulation results cannot represent instantaneous AT change, the value of the simulated and the measured values at the same position can be used for verification.

Pearson’s correlation coefficient was between −1 and 1, meaning that there was no correlation between the measured and the simulated with the absolute value closer to 0. A stronger correlation is when the absolute value is closer to 1, so a Pearson’s value greater than 0.85 indicates a strong correlation. The predicted value was closer to the actual value as the smaller MAPE value. *R*^2^ ranged from 0 to 1, the larger of *R*^2^ (closer to 1), the better fit of the predicted model. The two sets of data can be said to be correlated when the *R^2^* value is larger than 0.6. IBM SPSS Statistics (25) and Origin (2018) were used for mathematical statistical analysis.

## 3. Results

### 3.1. Air Temperature

#### 3.1.1. Measured Air Temperature

As shown in Figure 3, the trends of the measured and the meteorological station were basically similar, i.e., an increase first and then decreasing; however, the AT of the meteorological station had a relatively strong fluctuation compared to the measured. The highest AT value of the sites appeared at 16:00, except for site 3, while the highest AT value of site 3 occurred at 15:00, close to 34.16 °C. The difference period of the highest AT among all the sites may be due to the influence of the spatial form around the measuring sites on the local thermal environment. Worth noting, the AT of the weather station before 11:00 was basically close to the four measuring sites; however, the AT data of the weather station after 11:00 were higher than the measured. This may be due to the urban heat waves contributing to air-temperature data from weather stations with the increasing sun radiation, because the weather station is located in urban high-density areas. Nonetheless, the trend of air temperature measured throughout the day in this study is still consistent with the weather stations.

#### 3.1.2. ENVI-Met Model Validation

The accuracy of ENVI-met was verified through the measured data from the four sites in DN village on 21 July 2018. Figure 4 is a scatter diagram of the measurement and simulation, and Table 5 illustrates the values of Pearson’s correlations and MAPE.

It can be seen that the overall *R*^2^ was 0.853 and the Pearson value 0.929, meaning that all the measured AT had a strong correlation with the simulation. The Pearson values of sites 1 and 2 were slightly lower than those of sites 3 and 4, and all Pearson coefficients were greater than 0.8, which indicates that the measured and simulated AT were strongly correlated. In addition, the MAPE ranged from 2.16% to 3.79%. According to previous studies, it is considered that the error between the actual measurement and the simulation can be ignored due to the value of MAPE being less than 5% [59,60]. Results showed that the simulated and measurement of AT had a strong correlation and the average percentage error between them could be ignored. Therefore, the ENVI-met model is valid in this study and can simulate the village microclimate environment.

#### 3.1.3. Changes in Air Temperature in Villages

The microclimate environments of CJ village and DN village were simulated through the same input meteorological parameters after verifying the accuracy of the ENVI-met. AT of 1.5 m on the ground at 9:00 and 13:00 was extracted from the simulation to analyze the outdoor thermal environment, including the AT data of W-E and S-N roads in each zone of nine equal zones. CJ village has only eight sets of data since CJ−1 zone has no roads, and DN village contains nine sets of data. Figure 5 illustrates the AT of W-E and S-N roads in the two villages, and it can be clearly seen that the AT of 13:00 is higher than that of 9:00. The AT of W-E road was slightly higher than that of the S-N road, but with little AT value.

The AT-difference values was calculated as illustrated in Figure 6, which is a box plot of the AT difference between W-E and S-N roads in both DN village and CJ village. Clearly, the values at 9:00 was higher than those at 13:00 in DN village. In terms of CJ village, the AT difference range at 9:00 was higher than that at 13:00; indicating that the temperature of the W-E road was higher than that of the S-N road, especially in the morning.

This may be due to the angle of solar radiation and the earth surface being small while the sun rises in the morning, and the orientation of the S-N road is basically perpendicular to the solar radiation, which makes it difficult for the solar radiation to reach the earth surface. The W-E road is parallel to the solar radiation, which makes the W-E road absorb more solar radiation than the S-N road. Therefore, the long-wave radiation reflected from the ground into the air of W-E was greater than that of S-N, revealing that the AT of W-E was higher than that of S-N. However, the angle of solar radiation at 13:00 was basically perpendicular to the ground, and the W-E road and the S-N road basically absorbed the same amount of solar radiation. As such, we determined that the AT of the W-E road was higher than that of the S-N road and the most significance AT difference occurred in the morning.

### 3.2. Multiple Regression Model Setup

In order to further explore the mechanism between the village spatial forms and the microclimate, the percentage of building coverage (PBC), the ratio of building height to road width (H/W), and wind speed were selected as independent variables. The AT of 1.5 m from the ground was the dependent variable, establishing a multiple regression model. Every equal zone divided by the method of three by three grids basically covers the whole village position, and the independent variable of PBC and H/W in each zone could represent the morphology of villages in the northwest region, because the study site of this paper is the typical village in the northwest region.

#### 3.2.1. Model Results

*R*^2^ is the percentage of change in the dependent variable, with closer to 1 indicating a more meaningful regression model, which is called coefficient of determination. F value is the significance test to determine whether the equation is meaningful or not, which is called the variance test. T value is a stepwise test to determine whether the regression coefficient is meaningful. VIF (variance inflation factor) is used to assess the level of collinearity, 1–10 is considered acceptable, and a value of >10 indicates that the model has significant multicollinearity problems. From the summary of the model (Table 6), we can see that the fit of the model was relatively good, as the *R*^2^ was 0.696, adjusted *R*^2^ 0.630, and significance F was 0.001. Furthermore, values of VIF reveal that the model had no serious multicollinearity problems. Thus, the independent variables of the model could accurately predict the dependent variable changes.

The original equations required transformation into standardized regression equations with a standard coefficient, in order to compare the relative importance of predictors. The value of standardized regression coefficients is generally between −1 and 1, and predictors have more influence on dependent variables with increasing absolute values of the standardized regression coefficient. The absolute standardized regression coefficient values of wind speed (−0.459) and H/W (0.487) were relatively similar, also larger than that of PBC (0.154). These findings suggest that the independent variable of PBC had the lowest contribution to the variation of the dependent variable temperature in the model, while the H/W was the most significant factor. The multiple regression model was established:AT = −0.334 ∗ Wind speed + 0.447 ∗ H/W + 0.002 ∗ PBC + 32.843(2)

#### 3.2.2. Normality

Figure 7 suggests that the values of the sample observations roughly agreed with the hypothesis of a normal distribution, due to the frequency distribution of the standardized residuals matching the distribution of the normal curves. The observations were close to the hypothesis of a normal distribution, as the cumulative feasibility points for the values of standardized residuals were mostly distributed evenly among both sides of the straight line. Therefore, this can prove that the residual distribution followed a normal distribution through comparison (i.e., the sample observations exhibited normality).

#### 3.2.3. The Relationship between H/W and AT

In order to further analyze the coupling relationship between H/W and village microclimate environment, 50 AT simulation data at 13:00 in CJ village and DN village was obtained, which were evenly distributed in the villages, while calculating the corresponding H/W.

Figure 8 shows the scatterplot of H/W and AT. Obviously, both of them are negatively correlated and the correlation coefficient is −0.65, indicating a relatively strong correlation. Although the H/W data were not representative of everywhere in the village, they were evenly distributed in the village and could describe the basic village form. According to the box plot of H/W (Figure 9), except for abnormal data, the maximum value of H/W was 0.93, the minimum value 0.52, and the average value 0.76. Therefore, we believe that H/W and air temperature had a relatively strong negative correlation (Pearson −0.65) when H/W was between 0.52 and 0.93, providing data support for rural planning.

### 3.3. PMV

Air temperature cannot represent human thermal sensation, so the thermal comfort index (PMV) was used to evaluate the outdoor thermal comfort, as shown in Figure 10. The PMV data were derived at 1.5 m from the ground on the W-E and S-N roads in every equal area zone (from site 1 to site 9). The PMV performance of the two villages increased first and then decreased. The very hot period appeared from 11:00 to 18:00. After 21:00, both the villages were basically slightly warm. It is worth noting that the PMV of sites 7 and 8 were relatively low, especially DN village from 10:00 to 18:00 and CJ village from 9:00 to 18:00. Although these time periods are different, they are both in the daytime.

Figure 11 shows the box plots of PMVs from 8:00 to 23:00 in the two villages. Obviously, the minimum PMV values of sites 1–9 of DN village and CJ village are basically the same, but the maximum is different. More specifically, average PMV of site 7 and site 8 of DN village was about 3.19, while the mean PMV of the other seven sites was about 3.5, with a difference of 0.31. For CJ village, the mean PMV of site 7 and site 8 was about 3.3, while the others were about 3.5, a difference of 0.2. This means that the average PMVs of the two villages were relatively close, but the PMVs of site 7 and site 8 were smaller than the other seven areas.

In general, results of PMV evaluation revealed that site 7 and site 8 were the smallest PMV areas of the whole village; however, both site 7 and site 8 were located at the leeward edge of the village construction area, as this current research used the fixed northeast wind, and thus, we can trust that PMV at the edge of the downwind area is the smallest of the overall village. Although the PMV difference was not very significant, we used the PMV index to evaluate the thermal comfort of the village, which provides a basis for rural planning.

## 4. Discussion

Rural revitalization was a central policy of the Chinese government in 2021, aiming to strengthen rural economic construction. Meanwhile, the living standards between rural and urban would be gradually narrowed through rural revitalization. The implementation of rural revitalization has been studied from different perspectives through investigating villages in Guangdong Province [31], revealing that the complexity and unique features should be considered while implementing the rural revitalization strategy. Refs. [61,62] indicate that the improvement in land transfer laws and land market is of great significance to rural vitalization. Rural revitalization strategies based on the spatiotemporal pattern of rural production–living–ecological laws should be formulated in the future [63]. Moreover, Ref. [64] proposed measures to improve cultivated land to support rural revitalization to explore the primary limiting factors of agricultural land quality. Villages are sites of tourism, leisure, and consumption instead of agricultural commodity production in the postmodern world, and rural tourism has been recognized as a key approach to achieve rural revitalization. Ref. [65] also reveals that material, social, and spiritual perspectives are effective pathways for successful village revitalization. It is suggested that local governments strengthen rural tourism activities to stimulate rural revitalization by analyzing the driving factors of rural communities’ spatial and social evolution from the perspective of tourism [66]. This paper provides strategies for rural tourism planning to support rural revitalization through investigating village microclimate characteristics.

Through investigating the village microclimate characteristics, it was found that the air temperature on the S-N road is lower than that on the W-E, especially in the morning. This not only supplements the gap of the village microclimate, but also provides a reference for village design. For example, priority is given to designing morning markets on S-N roads, because lower air temperatures can create a better outdoor environment and attract more tourists, as well as focusing on the planning of facilities with a large flow of population on S-N roads, such as food streets, entertainment streets, etc. In addition, the evaluation results of outdoor thermal comfort revealed that site 7 and site 8 were the zones with the lowest PMV values in the whole village. Although the difference in PMV value was not very obvious, it evaluated the thermal comfort of the village and also divides different areas with PMV values. We can determine that the leeward area of the village with the constant wind direction is a relatively comfortable area. Although instantaneous wind cannot be simulated, the results could represent the thermal environment of the village with dominant wind, which is the highest wind frequency during a whole year.

In terms of the villages developing rural tourism, the land space of village construction area should be divided by function according to the division of outdoor thermal comfort based on this research. For example, the leeward area of the local constant wind direction in the village should be focused on hotel accommodation, because the zone is relatively comfortable. Facilities such as parking lots or green landscapes could be arranged in the windward area due to the impact of the thermal environment on people’s comfort being relatively weak in parking lots, as people stay a short time. Moreover, green landscapes have obvious cooling effect on downwind areas. Our results provide a basis for village land use, which has guiding significance for rural tourism also promoting rural revitalization.

The regression model also shows that H/W is the main factor affecting the thermal environment compared to PBC and wind speed. This may be different from cities. The infinite expansion of the impervious surface is the root cause of urban heat waves, and the secondary radiation generated between buildings is also an important factor leading to increasing air temperatures. The built-up area in rural areas is significantly smaller than that in cities and the building height is basically two stories, meaning that increasing of air temperature caused by impervious surfaces and building height is not significant. Thus, optimizing the H/W is an effective strategy to improve the outdoor thermal environment in villages, supported by H/W having a relatively strong negative correlation with air temperature, while H/W was between 0.52 and 0.93. These results provide data support for rural planning from the perspective of the outdoor thermal environment.

## 5. Conclusions

Field measurement and computer simulation were used to investigate the microclimate of villages in the northwest during summer. Three-by-three grids were developed to evaluate average microclimate environment, and PMV was used to evaluate outdoor thermal comfort. Main results are as follows:(1)ENVI-met can effectively predict the village microclimate environment due to the correlation coefficients being greater than 0.8 and the MAPE ranging from 2.16% to 3.79%.(2)Air temperature on the W-E roads is higher than that of the S-N, especially in the morning, supplementing the gap in village microclimate and providing guidance for rural land use.(3)The regression model showed that the H/W was the main factor affecting village microclimate compared with wind speed and PBC, revealing that optimizing H/W is an essential way to improve the rural outdoor thermal environment. Moreover, this study also found that H/W and air temperature had a relatively strong negative correlation (Pearson −0.65) when H/W was between 0.52 and 0.93.(4)The downwind area of the village based on the local dominant wind is a relatively comfortable area based on assessment of outdoor thermal comfort.

However, our research results are limited, only applying to the villages located in plains. Because the microclimate environment is significantly affected by surrounding topography, such as mountains or hills, the study results cannot accurately assess the influence of topography. Through discussion, some opinions are provided to local rural tourism, which will contribute to rural revitalization:(1)Rural policy makers should give priority to commercial spaces with a large flow population on S-N roads, such as morning markets, food streets, etc., mainly as the air temperature on S-N roads is lower than that of E-W, especially in the morning, and the lower air temperature could create a more comfortable living environment.(2)It is recommended that densely populated facilities be designed in the downwind area of a village’s built-up area based on prevailing winds, planning facilities with fewer personnel in the upwind area, because the downwind area is a relatively comfortable area compared to the upwind area. For example, give priority to hotels in the downwind area, and plan parking lots or green landscapes in the upwind area.

Even through this research achieved some results, this is only the preliminary stage, and it still needs improvement. In future research, we need to investigate more villages and obtain more microclimate data. The microclimate characteristics of villages with different terrains should be considered. Moreover, the thermal effect of the wall was ignored when ENVI-met was used to simulate the thermal environment. The secondary radiation and shadow generated by the wall have a significant impact on the air temperature. The thermal environment effect of the simulated wall will be considered to reduce errors.

## Figures and Tables

**Figure 1 ijerph-19-08310-f001:**
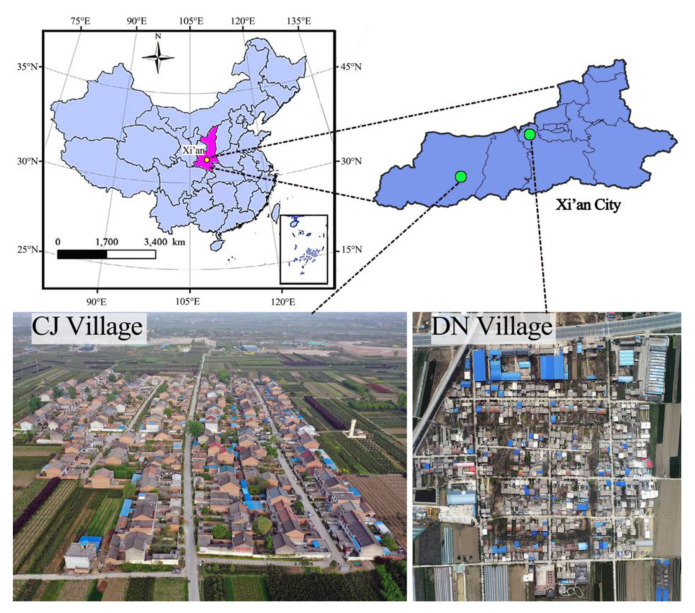
Location of the study site (images of Cai Jia Village and Da Nihe Village were photographed by drone).

**Figure 2 ijerph-19-08310-f002:**
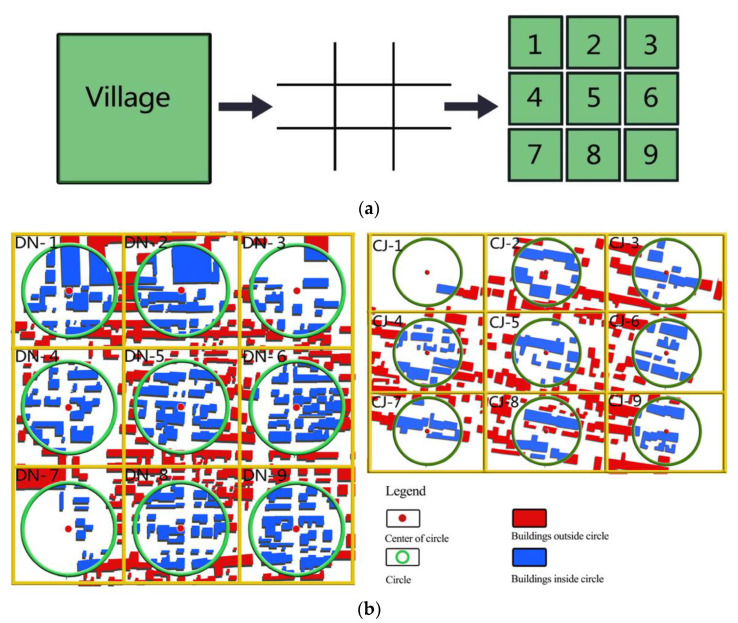
CJ village and DN village were divided into nine equal zones with the method of three-by-three grids: (**a**) method; (**b**) schematic diagram of the calculation method of PBC.

**Figure 3 ijerph-19-08310-f003:**
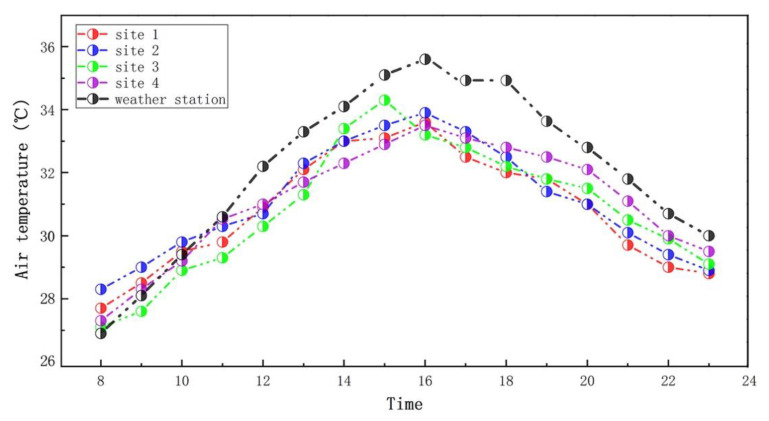
Air-temperature trends between measurement and weather station from 8:00 to 23:00 on 21 July 2018.

**Figure 4 ijerph-19-08310-f004:**
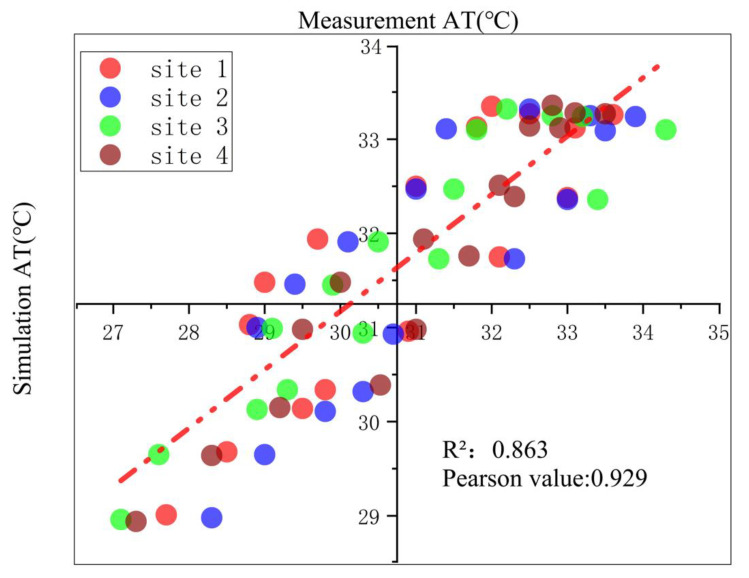
Correlation between the measurement (x-axis) and simulation (y-axis) data of air temperature on 21 July 2018. The AT of simulated and measured of the scattered points falling on the line are averaged; the red dashed line is the 1:1 line.

**Figure 5 ijerph-19-08310-f005:**
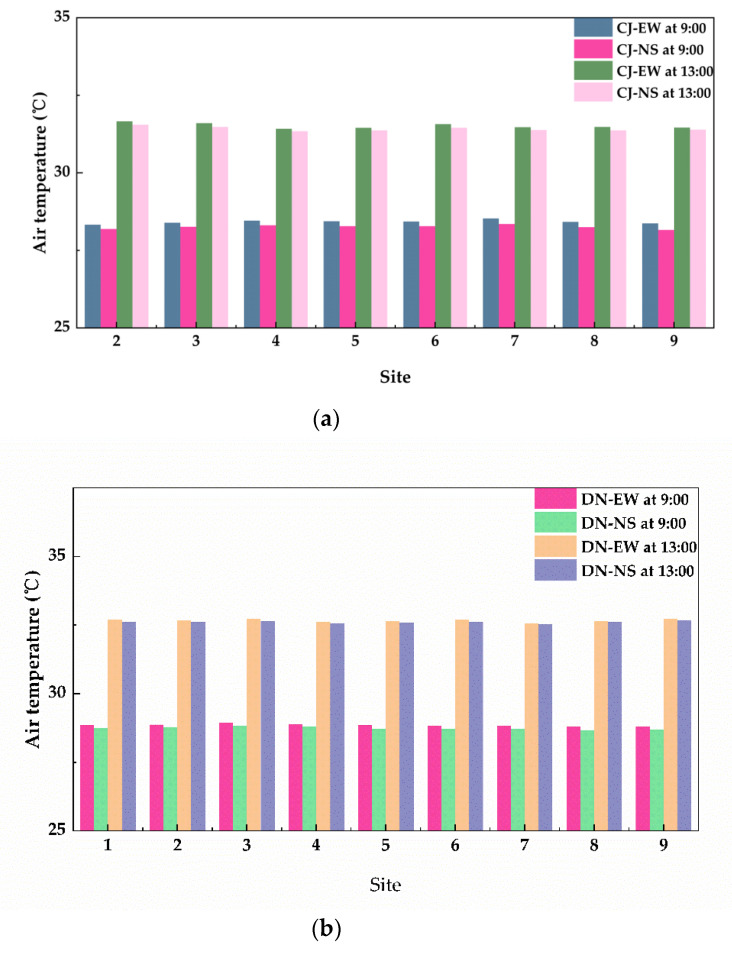
Comparison of the air temperature between W-E road and S-N road at 1.5 m from the ground at 9:00 (morning) and 13:00 (afternoon) on 21 July 2018, (**a**) CJ village, (**b**) DN village.

**Figure 6 ijerph-19-08310-f006:**
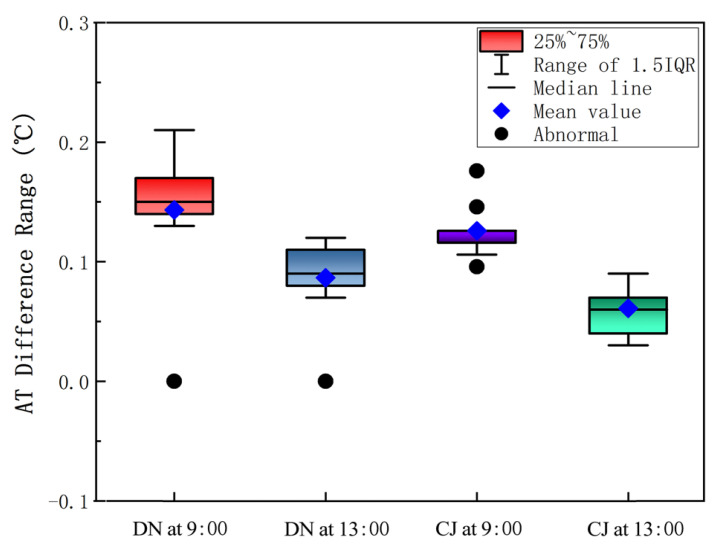
Box plot of air-temperature difference between W-E road and S-N road at 1.5 m from the ground at 9:00 (morning) and 13:00 (afternoon) on 21 July 2018.

**Figure 7 ijerph-19-08310-f007:**
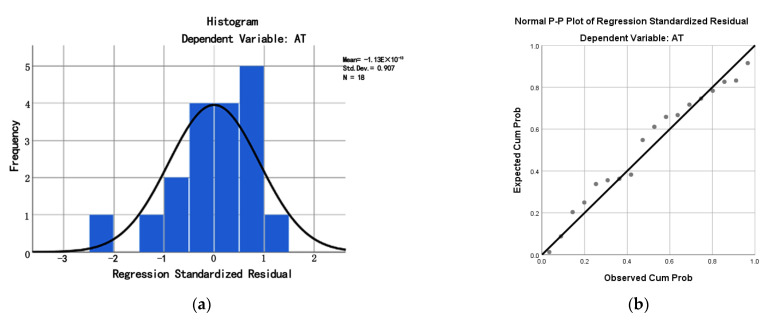
Standardized residual histograms (**a**) and P–P diagrams (**b**) of the regression model.

**Figure 8 ijerph-19-08310-f008:**
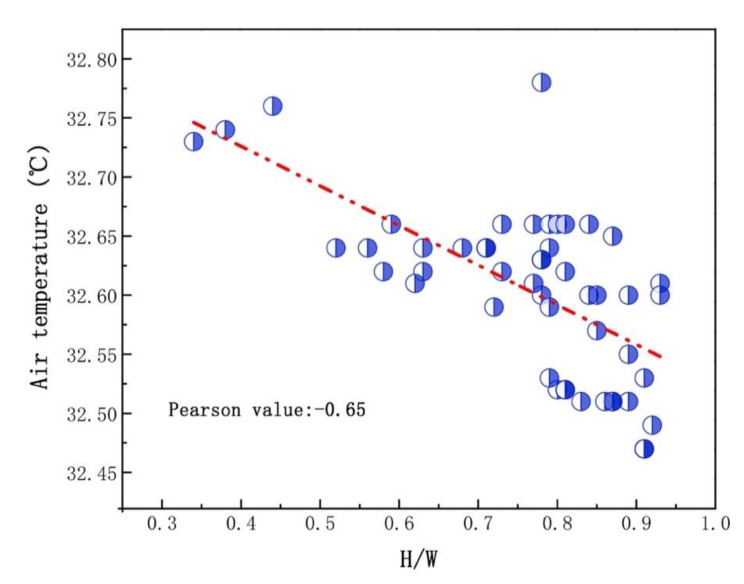
Scatterplot of H/W (the ratio of height and road width) and AT (air temperature) at 13:00 on a hot summer day (21 July 2018).

**Figure 9 ijerph-19-08310-f009:**
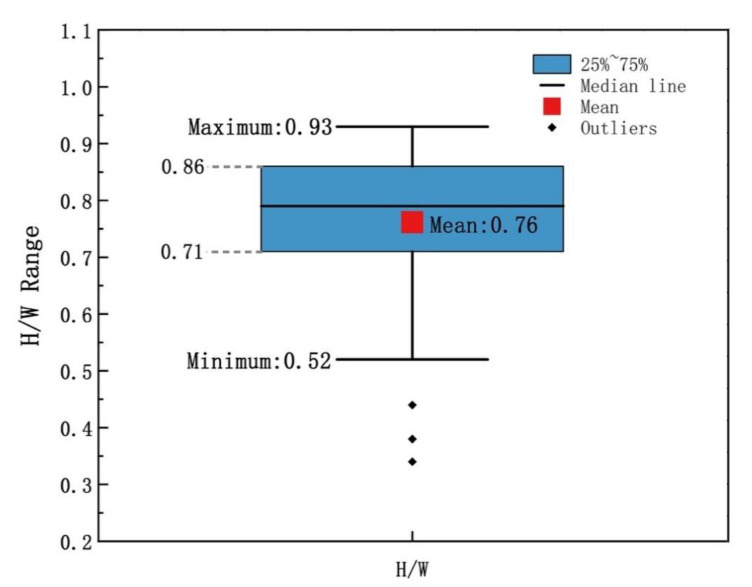
Box plot of H/W (the ratio of height and road width).

**Figure 10 ijerph-19-08310-f010:**
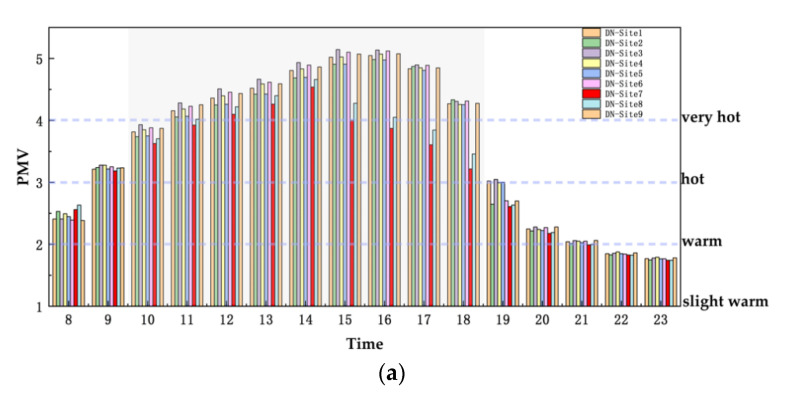
PMV histograms of the villages from 8:00 to 23:00 on 21 July 2018: (**a**) DN village, (**b**) CJ village.

**Figure 11 ijerph-19-08310-f011:**
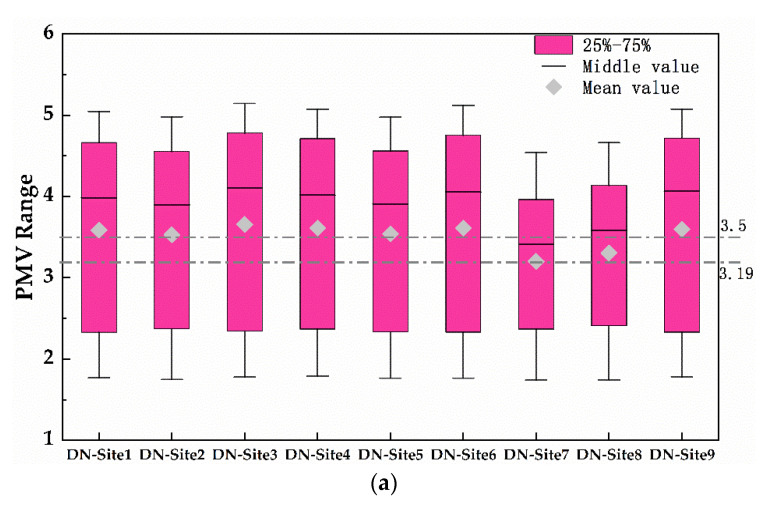
Box plot of mean PET at nine equal zones: (**a**) DN village, (**b**) CJ village.

**Table 1 ijerph-19-08310-t001:** Main thermal attributes of buildings and surfaces in ENVI-met model.

Parameter Type	Parameter Name	Setting Value
Buildings	Roof	Thickness (m): 0.30 Albedo (%): 0.50 emissivity (%): 0.90 Heat capacity [L/(m^3^·K)·10^−6^]: 1300 Heat conductivity (W/m·K): 0.84 Density (kg/m^3^): 1900 Roughness (m): 0.02
	Wall	Thickness (m): 0.30 Albedo (%): 0.30 Emissivity (%): 0.90 Heat capacity [L/(m^3^·K)·10^−6^]: 1050 Heat conductivity (W/m·K): 0.81 Density (kg/m^3^): 1800 Roughness (m): 0.02
Artificial surfaces	Concrete	Thickness (m): 0.30 Heat capacity [L/(m^3^·K)·10^−6^]: 2.25 Heat conductivity (W/m·K): 1.05 Roughness (m): 0.01 Albedo (%): 0.20 Emissivity (%): 0.90
Natural surfaces	Soil	Heat capacity [L/(m^3^·K)·10^−6^]: 1.21 Heat conductivity (W/m·K): 0.00 Roughness (m): 0.02 Albedo (%): 0.20 Emissivity (%): 0.96

**Table 2 ijerph-19-08310-t002:** Basic meteorological parameters and values during the simulation.

Settings	Parameter	Value
Simulation settings	Total simulation time	36 h
	Output time interval	60 min
	Number of nested grids	10
Initial parameter settings	Simulation start date	21 July 2018
	Simulation start time	07:00
	Initial temperature	26.0 °C
	Wind speed at 10 m	2.3 m/s
	Wind direction at 10 m	60° N-E
	Relative humidity at 2 m	40%
	Specific humidity at 2500 m	7 g/kg

**Table 3 ijerph-19-08310-t003:** Personal human parameters (according to “Standard human” ISO7730).

Body Parameters	Clothing Parameters	Person’s Metabolism
Age (year): 35 Gender: Male Weight (kg): 75.00 Height (m): 1.75 Surface area (DuBois area): 1.91 m^2^	Static clothing insulation (clo): 0.90	Total metabolic rate (W): 164.49 (=86.21 W/m^2^) (met): 1.48

**Table 4 ijerph-19-08310-t004:** Percentage of building coverage (PBC) within a 150 m circle and the ratio between height and width (H/W) in two villages.

Number	CJ Village		DN Village	
	PBC	H/W	PBC	H/W
Site 1	9.11%	0.59	48.56%	0.62
Site 2	42.56%	0.68	49.38%	0.69
Site 3	31.11%	0.53	25.62%	0.66
Site 4	41.58%	0.65	55.31%	0.63
Site 5	26.73%	0.68	47.43%	0.62
Site 6	33.72%	0.61	56.82%	0.73
Site 7	30.26%	0.68	17.82%	0.71
Site 8	40.73%	0.45	40.94%	0.38
Site 9	33.44%	0.54	46.47%	0.65

**Table 5 ijerph-19-08310-t005:** Pearson correlation coefficient and mean absolute percentage error (MAPE) of observed and simulated data.

Location	Site 1	Site 2	Site 3	Site 4
Pearson	0.856 **	0.843 **	0.929 **	0.959 **
MAPE	3.53%	2.91%	3.79%	2.16%

Correlation is significant at α = 0.05 level; ** correlation is significant at α = 0.01.

**Table 6 ijerph-19-08310-t006:** Model summary.

*R* ^2^	Adjusted *R*^2^		Std. Error of the Estimate		F	Sig. F
0.696	0.630		0.09042		10.660	0.001
Independent variable	Unstandardized Coefficients		Standardized Coefficients	t	Sig.	Collinearity Statistics
	B	Std. Error	Beta			VIF
Constant	32.843	0.287		114.243	0.000	
Wind speed	−0.334	0.128	−0.459	−2.613	0.020	1.419
H/W	0.447	0.149	0.487	3.006	0.009	1.208
PBC	0.002	0.002	0.154	0.950	0.358	1.214

Predictors: (constant), PBC, H/W, wind speed; dependent variable: AT (air temperature).

## Data Availability

Not applicable.

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
