# Peer review of "Supporting Design to Develop Rural Revitalization through Investigating Village Microclimate Environments: A Case Study of Typical Villages in Northwest China"

_ijerph, 2022, doi:10.3390/ijerph19148310_

Round 1

Reviewer 1 Report

The manuscript deals with an investigation of the microclimate in two rural villages in northwest China simulated through an ENVI-met model. The aim is supporting village land use optimization and promoting rural revitalization. A method was proposed to explore the relationship between village microclimate, site morphology and outdoor thermal comfort.

Althought the topic of the study is timely and sound, and the investigation was presented in a well-structured manner, it is the opinion of the reviewer that the analysis has not been sufficiently developed as it should be expected in order to be published on a scientific journal as Int. J. Environ. Res. Public Health. Therefore, the Authors are strongly encouraged to perform additional experiments as well as to strenghten research method and discussion.

More in details, the following weaknesses and inaccuracy have been identified in the review process:

- the statistical analysis was poor (e.g., no proper normality test was used) and scarcely informative (e.g., further data mining is recommended);

- some basic thermodynamic concepts seem to be equivocated in the discussion (apparently a dependency of temperature from only wind and site morphology was claimed);

- the prediction accuracy of the ENVI-met model was assessed by an insufficient amount of data (i.e., 24 hours);

- overall, a solid scientific approach in data manipulation and interpretation seems to be lacking (e.g., inconsistent number of digits for various variables and parameters, missing abbreviations to interpret the formulas, not fully readable plot axes, unclear definition of significance to discuss differences among data).

Furthermore, it was sometimes difficult to follow the text and an extensive editing of English language and style is needed.

Author Response

请参阅附件。

Reviewer 2 Report

The study addresses the issue of climate change and how, from a local perspective, the analysis of various parameters associated with climate change can provide a better understanding of local environments to promote social and economic well-being. In this logic, addressing microclimates can constitute an effective revitalisation strategy for vulnerable rural territories.

The introductory section is quite explanatory but the theoretical and conceptual contributions are deficient, and improvements are recommended.  The methodological section notes that the study focuses on two Chinese villages in northeast China. This methodology is quite ambitious, however, aspects of the number of villages studied may lead to a misperception with only two local areas, I recommend the authors to specify the study area, make the necessary changes in the title of the manuscript or change the study to a case report which would fit well, I think the editor should consider the latter aspect. 

An important aspect that the authors do not mention is the origin of the computer programs used to make the simulations, they only mention, but do not describe, where they were designed and which country, if these programs were adjusted to the study area or if they made specific adjustments for the study area.

Figures 1 and 2 are of low quality, it is necessary to improve the figures presented considering the editorial standards of the journal. The research design is confusing, although the computational mechanisms used are mentioned they are not explained, at the beginning of the methodological section there is no mention of the research approach used, nor are the research methods mentioned, as well as the scope and limitations of the study, it is important to mention and explain these methodological elements of the research.

In the discussion section, the results obtained are not discussed in depth, there is no dynamic of contrast with similar studies, it is evident that this section requires substantial improvement.  Likewise, the conclusions section is deficient, the abundant information presented in the results is not considered in its entirety, which leads to biased conclusions. Both sections require substantial reflection and improvement.

The biobibliography used is poor and deficient; it is important to expand the references by considering the incorporation of updated bibliography.

Overall, the manuscript presented by the authors is an interesting and important study in the context of climate change and human well-being, but the authors should address the suggestions made to strengthen the study so that it has a better chance of being published.

Reviewer 3 Report

The paper regards a methodology for the rural revitalization in Northwest of China through the analysis of the microclimate. The importance of the topic is fundamental in a climate change scenario. However, there are some flaws that need to be adjusted, especially regarding the evaluation of the microclimate.

Introduction

The citation https://www.un.org/en/climatechange/what-is-climate-change does not sound as a very scientific source. I recommend replacing it with data from scientific papers

Climate change also affects indoor thermal comfort and energy consumption (you may see https://doi.org/10.3390/su131810315 which considers the effect of climate on buildings such as schools, which can be particularly critical) and the studies on adaptation to the climate (see the book “Adapting Buildings and Cities for Climate Change”)

Methodology

- Please move the "https://www.envi-met.com/, accessed on 21 February 2022" in the references and not directly in the text

- How long were the data monitored (24h?)? Please specify it in the methodology. Then, I suggest validating the ENVI-Met model on all the data acquired and not just on one day.

- I recommend providing a paragraph to describe the monitored parameters and not include it in the ENVI-met model.

- 2.3, please change in 2.3. Outdoor thermal comfort (not ourdoor)

- PMV: I don’t think that the PMV is a good index to evaluate outdoor thermal comfort, as the scientific community is already discussing its applicability indoors (see https://doi.org/10.1016/S0378-7788(02)00018-Xhttps://doi.org/10.1016/j.buildenv.2019.01.055). Its application to outdoor environments is often inaccurate as the conditions are more dynamic, and PMV tends to overestimate the thermal sensation towards the warmer end of the scale in hot climates and vice versa in cold climates (see 10.1109/EEEIC/ICPSEurope51590.2021.9584714,  https://doi.org/10.1016/S0378-7788(02)00017-8https://doi.org/10.1016/S0038-092X(00)00093-1). An index specifically developed for outdoors must be used to avoid errors in comfort assessment.

Results

It is confirmed what was previously noticed on the PMV index. Values of PMV reach even +5 which is out of the PMV range. I recommend using another index to assess outdoor comfort, first because outdoor conditions are not stationary, but also because it does not take into account possible adaptation.

Discussion

It should be improved. A discussion on the main findings of the paper should be added, as well as a comparison with previous studies. In the introduction, you should identify the main literature regarding rural revitalization and, in the discussion, you must compare your findings with others that can be found in the literature, to support your research. This is currently missing.

Conclusion

You should focus more on the research question. What is the importance of this study? Is it only the analysis of a case study? How can these findings be used to implement rural revitalization?

Round 2

Reviewer 1 Report

Dear Authors,

thank you for your kind reply to my comments.

You made a considerable effort to improve the manuscript, mostly in terms of English style. Nonetheless, I believe that methods and results still need further improvement in order to reach a sufficient quality to be published in IJERPH.

More in detail, for example:

- the statistical analysis was not substantially reinforced with new data nor elaboration;

- in order to evaluate the accuracy  of the ENVI-met model, I would expect at least one week from different season;

- some abbreviations are still missing (e.g., Equation 1) and an inconsistent number of digits (e.g., temperature graphs) was used.

As the topic is sound and timely, I would strongly encourage the Authors to run more experiments and to provide more solid scientific methods to reinforce the validity of the results.

Reviewer 2 Report

The authors have made an important effort to improve their manuscript, however, I am concerned that in the discussion section nothing is discussed, there the authors mention only one reference and with only one reference there is no discussion.  The deficiency of this section may be associated with the fact that the authors did not heed my recommendation to expand and update their bibliography.
